# Developmental Differences between Anthers of Diploid and Autotetraploid Rice at Meiosis

**DOI:** 10.3390/plants11131647

**Published:** 2022-06-22

**Authors:** Tianya Ku, Huihui Gu, Zishuang Li, Baoming Tian, Zhengqing Xie, Gongyao Shi, Weiwei Chen, Fang Wei, Gangqiang Cao

**Affiliations:** Henan International Joint Laboratory of Crop Gene Resources and Improvements, School of Agricultural Sciences, Zhengzhou University, Zhengzhou 450001, China; kxm0305@163.com (T.K.); hhgu@zzu.edu.cn (H.G.); lzs517@126.com (Z.L.); tianbm@zzu.edu.cn (B.T.); zqxie@zzu.edu.cn (Z.X.); shigy@zzu.edu.cn (G.S.); weiwei_chen15134@zzu.edu.cn (W.C.)

**Keywords:** autotetraploid, fertility, meiosis, polyploidy, proteomics, rice

## Abstract

Newly synthetic autotetraploid rice shows lower pollen fertility and seed setting rate relative to diploid rice, which hinders its domestication and breeding. In this study, cytological analysis showed that at meiosis I stage, an unbalanced segregation of homologous chromosomes, occurred as well as an early degeneration of tapetal cells in autotetraploid rice. We identified 941 differentially expressed proteins (DEPs) in anthers (meiosis I), including 489 upregulated and 452 downregulated proteins. The DEPs identified were related to post-translational modifications such as protein ubiquitination. These modifications are related to chromatin remodeling and homologous recombination abnormalities during meiosis. In addition, proteins related to the pentose phosphate pathway (BGIOSGA016558, BGIOSGA022166, and BGIOSGA028743) were downregulated. This may be related to the failure of autotetraploid rice to provide the energy needed for cell development after polyploidization, which then ultimately leads to the early degradation of the tapetum. Moreover, we also found that proteins (BGIOSGA017346 and BGIOSGA027368) related to glutenin degradation were upregulated, indicating that a large loss of glutenin cannot provide nutrition for the development of tapetum, resulting in early degradation of tapetum. Taken together, these evidences may help to understand the differences in anther development between diploid and autotetraploid rice during meiosis.

## 1. Introduction

Polyploidization promotes plant evolution and the formation of new species, which thereby contributes greatly to plant biodiversity [1]. Most ancestral angiosperms experienced one or more chromosome doubling events, which is an important mechanism causing speciation in eukaryotes [2,3]. Compared to their predecessors, autopolyploid plants were taller, had thicker stems, displayed thickened and enlarged leaves, and grew over a longer period [4,5]. However, polyploid plants also showed deficiencies in development time, fertility, and seed setting rate. The doubling of the genome and/or new combinations of different chromosomes can improve the ability of polyploids to resist stress and adapt to environmental change [6]. Since the first development of autotetraploid rice in 1933, there has been considerable research on different forms of autopolyploid rice [7]. In particular, the breeding of PMeS (polyploid meiosis stability) tetraploid rice lines and neo-tetraploid rice lines resolved the low seed setting rate problem that was an issue with many types of autopolyploid rice [8]. Moreover, we believe that polyploid rice breeding is an attractive prospect for improving crop yield in the future [1,9].

Compared to diploid rice, autotetraploid rice has higher-quality agronomic traits, which have attracted considerable research attention [10,11]. For example, the height, flag leaf length, and flag leaf width of autotetraploid rice are all significantly different from diploid rice [12]. Before the 1960s, research mainly focused on the genetic background of rice, as well as hybridization between cultivated rice and wild rice and the evolutionary relationships among rice species [13]. In recent decades, theoretical research in polyploid rice breeding has also been conducted on topics such as artificial induction, the biological and reproductive characteristics of different polyploid rice varieties, rice genome dynamics during polyploidization, and evidence of distant hybridization [14,15]. Furthermore, scientists have also studied the ultrastructures involved in microspore formation in autotetraploid rice by transmission electron microscopy [16]. In one study, scientists explored the mechanisms involved in the decline of seed setting rate in autotetraploid rice by examining both autotetraploid rice and its interspecific hybrids [17,18]. These studies have laid the foundation for successful polyploid rice production in the future. 

However, despite these successes, the biological characteristics of autotetraploid rice remain understudied. At present, research on autotetraploid rice mainly focuses on fertility [19]. At the same time, considerable research has also focused on the cellular development of diploid rice. Studies have shown that the cytological mechanisms responsible for pollen sterility in diploid rice include the following: meiosis anomalies [20], the abnormal development of the tapetum [21], the abnormal vacuolation of microspores [22], and abnormal germ cell wall formation [23]. Autotetraploid rice was obtained by doubling the genome of diploid rice. Despite its potential in improving crop yield, its seed setting rate is low. During pollen development in autotetraploid rice, abnormal chromosome behaviors may have had direct or indirect effects on fertility. Moreover, infertility may occur at various stages, including the sporogenous cell stage, the uninucleate microspore stage, and the binucleate microspore stage [24]. In addition, there is a correlation between the normal development of both male and female gametophytes and their fertility in autotetraploid rice [25]. This is because the development of the plant embryo sac is directly related to fertility and seed setting rate. By studying polyploid rice using whole staining transparent laser scanning confocal microscopy (WCLSM), investigation of embryo sacs revealed that during the mitosis of functional megaspores, some egg cells degraded during differentiation in esd1 (embryo sac development 1) mutants, thereby hindering subsequent fertilization process and reducing seed setting rate [26].

Compared with diploid cells, autotetraploids may differ not only in cellular development but also in gene or protein expression levels due to chromosome doubling, such as anthers. For this reason, conducting research on autotetraploid cells is complex [27]. There have been many reports on pollen abortion and the abnormal development of pollen mother cells, but to date there are no general conclusions. In addition, it is unknown whether abnormal cell development will lead to infertility [28,29].

After rice polyploidization, the expression level of proteins in vivo will also change. These changes are related to both physiological conditions and variation in traits [30]. A large number of differentially expressed genes in vivo will affect the subsequent development of plants, and may affect both the transcription and translation processes [31]. Proteins have the most direct impact on organisms. Therefore, proteomics methods designed to quantitatively detect changes in protein expression are key for the exploration of the effects of rice polyploidization [32].

Studies have shown that autotetraploid rice shows reduced pollen activity, which may be related to the increase of the size of the rice genome, but changes in plant genome size do not always lead to reduced fertility. Moreover, rice anther development—especially in the early meiosis stage—is of great significance for sexual reproduction. The expression levels of anther proteins in autotetraploid rice vary with chromosome doubling during different developmental stages [33]. By identifying the differentially expressed proteins (DEPs) between the anthers of diploid and autotetraploid rice at early stages of development, and by observing differences in cellular and agronomic traits, we determined which protein regulation mechanisms were associated with infertility in autotetraploid rice when the genome size increases. These results will therefore provide a theoretical basis for understanding the impact of polyploidization on autotetraploid rice [19,34].

## 2. Results

### 2.1. Cellular Analysis of the Autotetraploid Rice

We examined polyploid rice by flow cytometry to verify that polyploidization had taken place, then observed and recorded the phenotype. The infertility of autotetraploid rice was observed by examining pollen grains. In addition, we explored the reasons for the low fertility of autotetraploid rice by observing the outer and inner walls of the anthers at meiosis I. First, our flow cytometry results showed that the autotetraploid rice had a relatively stable ploidy level throughout reproductive development compared to the diploid rice (Appendix A). In addition, chromosome counting showed that the chromosome number of diploid rice was 2x = 24 (Appendix A), and that of the autotetraploid rice was 4x = 48 (Appendix A). We also found that the anthers of the autotetraploid rice were longer than those of the diploid rice, and their color was lighter (Appendix A). These findings were consistent with polyploidization in angiosperms. Second, we found that the leaf length of the autotetraploid rice was longer than that of the diploid rice, that the leaves of the autotetraploid were thicker and wider, and that the autotetraploid rice plant was, in general, stronger and larger (Appendix A). Furthermore, we measured anther size, which provided a basis for sampling accuracy in subsequent proteomic analyses (Appendix A). We also observed pollen grains at meiosis I stage by microscope. We found that in the same unit area (the pollen grains of diploid rice were used as control), the autotetraploid rice showed more abortive pollen grains (Figure 1A,B), and the number of pollen grains it produced was 68.4% of the number produced by the diploid (Figure 1C). To further verify the accuracy of our Alexander staining results, the pollen grains from the two plant types were stained with iodine–potassium iodide. These results also showed that the number of abnormal pollen grains in the autotetraploid rice was greater than in the diploid rice (Figure 1D,E). 

Next, we measured the chlorophyll content of new leaves of diploid and autotetraploid rice. The chlorophyll content of autotetraploid rice was lower than that of diploid rice, suggesting that the function of proteins related to chlorophyll synthesis in chloroplasts may be affected in the autotetraploid (Figure 1F). 

We then explored whether the development of anthers at meiosis I was affected by polyploidization thus far by examining anthers using scanning electron microscopy (HITACHI SU3500). We applied the anther evenly to the conductive tape for observation. Our results showed that, whether diploid rice or autotetraploid rice, the structure of anther cuticle in meiosis I stage showed similar network structure, arranged regularly and developed well (Figure 1G,H). Subsequently, to explore whether the inner wall of the autotetraploid rice anthers was abnormally affected by polyploidization during meiosis I, the paraffin section technique was used to examine cross-sections of anthers at the early stage of their development. Our results showed that the tapetum was obviously degraded earlier in autotetraploid rice at the prophase of meiosis I (Figure 2). Early tapetum degradation may have direct and/or indirect effects on pollen viability in autotetraploids, resulting in reduced fertility.

Next, we investigated the processes involved in the diploid and autotetraploid rice cells at meiosis (Figure 3A). We found that most of the meiosis processes in both rice types were normal, but during meiosis I, some cells showed abnormalities during chromosome separation. In the case of normal chromosome separation, both ends of the cells are 24 chromosomes, but in autotetraploid rice chromosome unequal separation occurs, such as some cells having 22 chromosomes at one end, two less than normal cells. We divide chromosome segregation into six types: I (24/24); II (23/25, 22/26, 21/27); III (20/28, 19/29, 18/30); IV (17/31, 16/32, 15/33); V (14/34,13/35,12/36); VI (11/37,10/38). Of these, the most abnormal separation ratio is II (23/25, 22/26, 21/27), as shown in Figure 3B. Therefore, we conclude that the meiosis process of autotetraploid rice was affected, and that abnormalities during homologous chromosome segregation might lead to infertility. In addition, due to the abnormal degradation of tapetum, nutrients during development cannot be provided, resulting in abnormal spindle assembly, and, finally, abnormal chromosome separation occurs. We then performed proteomic analyses to explain cellular results.

### 2.2. Post-Translational Modifications and Cell Cycle Regulation Are Related to Abnormal Degradation of Tapetum in Autotetraploid Rice

According to the cytological results, the tapetum of autotetraploid rice developed abnormally at meiosis I stage. To investigate the mechanisms responsible for abnormal degradation of tapetum in autotetraploid rice, proteomic analysis was performed on the early anthers (meiosis I). DEPs (differentially expressed proteins) were defined as those proteins with a fold change (FC) = 1.2 when the false discovery rate was <0.05. A total of 941 DEPs were identified between diploid and autotetraploid rice. Of these, 489 were upregulated and 452 were downregulated (Figure 4). Using the KOG (Clusters of Orthologous Groups of proteins) database, the identified DEGs were annotated and divided into 25 categories (Figure 5A). We found that the “post-translational modification, protein turnover, and partner” annotations were very prominent. This suggests that post-transcriptional regulation may play an important role in pollen sterility in autotetraploid rice. In addition, we also found DEPs classified into three categories related to “cell cycle control, division, chromosome partition”, “replication, recombination, and repair”, and “chromosome structure”; among them, cell cycle regulation is essential for vacuolate microspores. Therefore, we use a Venn diagram to illustrate the relationship between post-translational modification and these items (Figure 5B). The results showed that *BGIOSGA004246*, *BGIOSGA019809*, *BGIOSGA003857*, and *BGIOSGA028039*, genes that regulate PP2A and CDKs, were found to be significantly downregulated in autotetraploid rice. The lack of expression and activity of PP2A and CDKs may lead to pollen abortion [35]. In addition, small ubiquitin-related modifier (SUMO), which is encoded by *BGIOSGA026036* and *BGIOSGA035304*, regulates a variety of cell pathways via protein modification. Notably, SUMO can regulate chromatin dynamics and play an important role in gene expression regulation [36]. Likewise, chromatin remodeling factor can promote DNA repair or transcriptional regulation via ubiquitination, which is an effective nucleosome E3-ubiquitin ligase [37]. Therefore, ubiquitination is also closely related to chromatin remodeling. Abnormal ubiquitination may also lead to abnormal chromatin, thereby affecting the fertility of autotetraploid rice.

Subsequently, items related to protein degradation were selected for KEGG (Kyoto Encyclopedia of Genes and Genomes) pathway analysis (Figure 5C). We focused on the ER-associated degradation (ERAD) pathway (Figure 5D) and protein ubiquitination modification (Figure 5E). The ERAD pathway is involved in endoplasmic reticulum stress-related protein degradation. Molecular chaperones, such as the heat shock protein Hsp70 family, play an important role in ERAD substrate recognition and screening. When *BGIOSGA007251*, *BGIOSGA008908*, *BGIOSGA017916*, and *BGIOSGA004771*, which control the synthesis of Hsp70 (Heat Shock Protein 70) family genes [38,39,40], are downregulated, this results in the failure of the recognition process. Therefore, we believe that the ERAD response of autotetraploid rice is not sensitive. This may lead to tapetum cells not growing normally, which results in apoptosis and, ultimately, premature degradation. 

Protein modification by ubiquitination is a common post-translational protein modification in eukaryotic cells [41]. The proteins encoded by *BGIOSGA002684* and *BGIOSGA035762* genes (ubiquitin binding enzyme E2) were found to be downregulated in autotetraploid samples. Ubiquitination is important for regulating protein degradation and is closely related to the ERAD pathway.

### 2.3. Abnormal Phosphopentose Pathway and Decrease of Glutenin in Autotetraploid Rice May Lead to Tapetum Degradation and Affect Pollen Fertility

To identify the potential mechanisms responsible for tapetum degradation and the corresponding reduction in pollen fertility, we performed KEGG analysis on the sugar-metabolism- and lipid-metabolism-related proteins identified by the KOG analysis results (Figure 5A). Using the KEGG database (Figure 6A), we found that our samples were enriched in proteins related to the photosynthesis (Figure 6B) and the pentose phosphate pathway (PPP) (Figure 6C). The *BGIOSGA016558*, *BGIOSGA022166*, and *BGIOSGA028743* genes involved in the pentose phosphate pathway were all downregulated; this meant that cells could not gain enough energy to make the tapetum develop normally. In addition, the series of intermediates and enzymes in the non-oxidative rearrangement phase are identical to many of the intermediates and enzymes from the photosynthetic Calvin cycle [42,43]. Thus, the pentose phosphate pathway can be linked to photosynthesis, where it can achieve interconversion between certain monosaccharides. This explains why autotetraploid rice has less chlorophyll in its chloroplasts compared to diploid rice.

In addition, amino-acid-metabolism-related processes were also selected from KOG analysis (Figure 5A) and eventually enriched for certain pathways (Figure 6D). One of the pathways worthy of our study is the catabolic process of glutamate (Figure 6E). The *BGIOSGA017346* and *BGIOSGA027368* genes, associated with key enzymes regulating the glutelins degradation process, are upregulated, leading to massive degradation of glutelins. The glutelins degradation process has a high nutritional value and is closely related to the development of the tapetum [44]. Therefore, the breakdown of glutelins may have led to the premature degradation of the tapetum, which could not absorb the nutrients required by the plant, making the plant unable to develop normally [45].

### 2.4. Verification of Proteomic Results by qRT-PCR (Real-Time Reverse Transcription-PCR)

Nine validating genes were selected for qRT-PCR (real-time reverse transcription-PCR), which we used to verify that the biological processes identified by our proteomic analyses were differentially expressed in the anthers of the autotetraploid and diploid rice. Primers used are listed in Appendix A. These results showed that the expression of most of the focal genes was consistent with the corresponding protein expression pattern (Figure 7). Only one gene expression pattern differed from the proteomic data. Taken together, these qRT-PCR results suggest close consistency between the transcriptional and protein levels with respect to the mechanisms involved in anther development in autotetraploid rice, as well as the involvement of post-transcriptional or post-translational modification processes.

## 3. Discussion

### 3.1. During Meiosis I, Chromosome Segregation and Degradation of Tapetum Were Abnormal in Autotetraploid Rice Compared with Diploid Rice

At present, research on autotetraploid rice mainly focuses on its low seed setting rate [46]. In general, plant polyploidization causes changes in cell components and cell structure, and can result in corresponding physiological disorders [47]. Studies have also shown that the infertility of autotetraploid rice pollen grains is not only caused by ploidy change, but is also affected indirectly by differences in gene and protein regulation.

Cell division requires the assembly of microtubule spindles to separate chromosomes correctly in mitosis and meiosis [48]. Separation is irreversible, and any error may lead to cell death. Cell assembly of microtubule spindles requires essential energy and nutrients. Microtubule spindle abnormalities can lead to unequal chromosome separation [49]. Studies have shown that some abnormal chromosome behaviors can lead to a decline in pollen grain viability, which is an important factor but may not always be the only one [47]. The results of this study showed that the abnormal segregation of homologous chromosomes occurred in autotetraploid rice during meiosis. In addition to abnormal chromosomal behavior, this experiment carried out cross-sectional observation of anthers at the meiosis stage, and found that in the meiosis I stage, the degree of degradation of the inner tapetum of autotetraploid anthers was not the same as in diploid rice. This early degradation leads to insufficient nutrients for cell development, such as microtubule spindle assembly. The wrong assembly will cause the unequal separation of chromosomes, thereby affecting plant development.

In addition, chlorophyll content measurements in autotetraploids were lower than in diploid rice, despite the larger width of their leaves. This phenomenon may be caused by ploidy changes, and it is likely that proteins or genes related to photosynthesis may be affected by this in plants, which in turn can cause a decrease in chlorophyll content.

### 3.2. Multi-Pathway Regulation of Degradation of Tapetum

Proteomics results showed that the upregulation of genes related to key enzymes regulating glutenin degradation led to a large amount of glutenin degradation, which cannot provide nutrients for plant growth and development and leads to the early degradation of tapetum [50]. In addition, ERAD pathway and ubiquitination play an important role in maintaining protein homeostasis.

Heat shock protein Hsp70 family plays an important role in ERAD substrate recognition and screening. The results of this study showed that the genes controlling Hsp70 family synthesis were downregulated, resulting in the recognition process being unable to proceed, causing endoplasmic reticulum pressure, thereby affecting the normal function of cells, including gene transcription, DNA damage repair, cell growth, apoptosis, and antigen presentation [51]. Ubiquitination involves covalently binding ubiquitin to lysine residues of substrate proteins under the synergistic effect of ubiquitin activating enzyme (E1), ubiquitin binding enzyme (E2), and ubiquitin ligase (E3). The ubiquitination system is the most important way to regulate protein degradation in eukaryotic cells, so it plays an important role in maintaining the dynamic balance of protein levels. In fact, the ubiquitination and phosphorylation of proteins have also been reported to play an important role in chromatin remodeling and homologous recombination during meiosis [52]. In addition, the ERAD pathway contains the ubiquitination process; some studies have shown that approximately one third of all synthesized proteins in a cell are channeled into the endoplasmic reticulum (ER) lumen or are incorporated into the ER membrane. Since all newly synthesized proteins enter the ER in an unfolded manner, folding must occur within the ER lumen or co-translationally, rendering misfolding events a serious threat. To prevent the accumulation of misfolded protein in the ER, proteins that fail the quality control undergo retrotranslocation into the cytosol where they proceed with ubiquitination and degradation [53]. The results showed that ERAD degradation pathway and ubiquitination-related proteins were abnormal, resulting in protein content disorder in plants, hindering normal cell development and perhaps causing abnormal degradation of tapetum.

The pentose phosphate pathway is a very complex process, which can not only provide energy in plants, but also in many other important functions [54]. The pentose phosphate pathway (PPP) is divided into an oxidative branch that makes pentose phosphates and a non-oxidative branch that consumes pentose phosphates, though the non-oxidative branch is considered reversible. A modified version of the non-oxidative branch is a critical component of the Calvin–Benson cycle that converts CO_2_ into sugar. The reaction sequence in the Calvin–Benson cycle is from triose phosphates to pentose phosphates, the opposite of the typical direction of the non-oxidative PPP. The photosynthetic direction is favored by replacing the transaldolase step of the normal non-oxidative PPP with a second aldolase reaction plus sedoheptulose-1,7-bisphosphatase [55]. In addition, Facchinello, Astone et al. demonstrated a role for the endothelial oxidative pentose phosphate pathway (oxPPP) in promoting vascular mural cell coverage and maturation during early development by regulating elastin expression. This mechanism establishes a critical role for oxPPP in the formation of the vascular system [56].

As an important storage protein, gluten is the main provider of nutrients. Studies have shown that the content and composition of glutenin are the key factors affecting the final quality of wheat [57]. Secondly, the synthesis, sorting, and processing of glutenin involve the synergistic effect of multiple regulatory factors. Identification and functional analysis of these regulatory factors are essential for rice quality improvement [58].

Therefore, we believe that autotetraploid rice cannot provide the energy required for cell development. After polyploidization, protein homeostasis is affected, and the development of tapetum is abnormal, resulting in a decrease in fertility.

## 4. Materials and Methods

### 4.1. Plant Material

The material used in this experiment was cultivated rice, and the control group was diploid indica rice 03–195. Autotetraploid rice was obtained by colchicine treatment of diploid rice. All plant material was planted in experimental fields at the College of Life Sciences, Zhengzhou University, Henan Province, China. The experimental and the control groups had different growth rates, so sampling times were inconsistent, but all tissues at sampling sites were of the same developmental stage. We selected rice anther tissue that was at an early stage of development for sampling. A total of 50 mg of each sample was weighed and snap-frozen in liquid nitrogen. Samples were then stored at −80 °C for subsequent proteomic research.

### 4.2. Ploidy Identification and Morphological Observation

The mature leaves of diploid rice and autotetraploid rice were used for ploidy detection. Detection instrument: Partec CyFlow Space; kit: Partec CyStain UV Precise P. Results determination method: The peak fluorescence intensity X-Mean (abscissa) was proportional to the cell DNA content, so the ploidy of each sample was determined by comparing X-Mean. For example, if the peak value of the control diploid sample is adjusted to 200 in the abscissa, the peak value of the haploid sample will appear at 100, and the peak value at 400 is tetraploid. The ordinate of the figure represents the number of cells, and the height of the peak reflects the difference in the number of cells. The mature leaves of autotetraploid rice and diploid rice were collected and placed into a fresh precooled dissociation solution (0.1 mol/L citric acid, 0.5% Tween 20, 4 °C). Leaves were quickly cut and filtered into a 1.5 mL centrifuge tube by 40 μm nylon mesh (Biologix, Shandong, China). After centrifugation for 30 s, 200 μL of the precooled dissociation solution was added to suspend the precipitate, and 600 μL staining solution (0.4 mol/L sodium dihydrogen phosphate, 0.1 mg/mL PI (propidium iodide), preserved at room temperature) was then added. After mixing, the DNA content of the samples was detected by flow cytometry, and the ploidy level of the sample to be tested was judged according to the peak position of a standard.

We made observations of both whole plants and anthers of autotetraploid and diploid rice. We recorded plant morphology and leaf size and shape. Images were captured by digital camera. Newly flowering rice anthers were spread on a sample table containing a conductive adhesive, were sprayed with an ion-sputtering apparatus, then observed and photographed with a Hitachi SU3500 scanning electron microscope.

### 4.3. Chromosome Counting

Rice flower buds were fixed in Carnoy’s solution (ethanol:glacial acetic acid = 3:1) for 8 h and stored in 70% ethanol. After observation under a stereomicroscope, the female gametophytes in the buds were removed and placed into an enzymolysis solution to fully hydrolyze them. The female gametophytes were digested in a wet box at 37 °C for 14 h, and were then transferred to water. The gametophytes were then completely smashed with an anatomical needle and large tissue fragments were removed. The cell homogenate was filtered using a 100 μm filter. Prepared slides were stained with PI for 10 min then photographed by an Olympus BX43 fluorescence microscope.

### 4.4. Pollen Viability Determination

Fresh rice flower buds were placed on clean slides, and an Alexander dye solution (Solarbio G3050 from Solarbio Science & Technology Co., Ltd., Beijing, China) was added dropwise. Sterilized tweezers were carefully taken out and used to smash the plant material, so that the pollen grains were released as evenly as possible into the dye solution [59]. Glass slides were covered, and filter paper was used to remove the dye solution. Pollen fertility was observed and photographed using an ordinary optical microscope. Pollen was counted using the staining results.

To increase the accuracy of our fertility statistics, we used an iodine–potassium iodide staining method for data filling. Five small blossoming flowers were randomly selected from each plant. Under the stereomicroscope, six anthers in each flower were carefully removed, and a 1% iodine–potassium iodide solution was added dropwise. After clipping the anthers, the residue was removed. To prevent bubbles, the cover-slip first covered the anther, and then the dye solution was removed by absorbent paper. Finally, specimens were photographed under an optical microscope.

### 4.5. Paraffin Section

First, rice anthers were fixed in Carnoy’s fixed solution (ethanol:glacial acetic acid = 3:1) for 3–5 h, then different concentrations of ethanol solution were prepared for gradient dehydration. Dehydrated samples were then placed into a solution of equal volumes ethanol and xylene for 10 min, and were subsequently transferred into pure xylene for 20 min to produce transparent samples. Second, we performed wax penetration and embedding. A solid wax block was removed from the embedding box and sliced on the slicer (Leica RM2231 from Leica microscopic system, Shanghai, China). Next, hematoxylin and eosin staining solutions were added for dyeing. After washing with pure water, dyed slides were soaked in pure water for 2 min. Finally, slides were dehydrated, transparent, and sealed. After drying, slides were observed under a common optical microscope and photographed [60].

### 4.6. Observation of Chromosome Behavior during Meiosis

The anthers of rice flower buds during meiosis were removed under an anatomical microscope, fixed overnight with Carnoy’s solution (ethanol:acetic acid, 3:1 *v*/*v*) at room temperature, then stored in 70% ethanol at 4 °C. Anthers were removed from the inflorescence using an anatomical needle under a stereo microscope. The samples were immersed in 100 μL of a pectinase–cellulase mixed solution (Solarbio C8270, Solarbio P818181 from Solebo Technology, Beijing, China) and placed into a wet box for 5–6 h at 37 °C. The prepared slides were stained with PI solution (40 g/mg) for 5 min, then observed under a fluorescence microscope [61].

### 4.7. Determination of Chloroplast Content

Fresh flag leaves of autotetraploid and diploid rice at early stage of development were collected. The leaves were then cut, placed into test tubes containing 10 mL chlorophyll extract, and placed at room temperature in the dark until the leaves became white. Taking the previously configured chlorophyll extract as a blank control, the absorbance values of the solution at wavelengths of 645 nm (chlorophyll a) and 663 nm (chlorophyll b) were measured [62].

### 4.8. Proteomic Analysis

Frozen anthers from preprepared rice were removed and ground in liquid nitrogen. Next, the anther lysates were ultrasonically treated on ice and the supernatant was precipitated with precooled TCA at −20 °C. Samples were then rinsed with acetone and the protein concentration was then determined. After being reduced at 37 °C for 60 min in 10 mM DTT (Sigma from Shanghai Sigma-Aldrich Trading Co., Ltd., Shanghai, China), protein samples were digested overnight. TMT labeling was performed using a 6-plex TMT kit as per the manufacturer’s instructions (Thermo Fisher from Thermo Fisher Technology Co., Ltd., Shanghai, China) [63]. Samples were then subjected to proteomic analysis by Shanghai Luming Biology. The mass spectrometry proteomics data have been deposited to the ProteomeXchange Consortium (http://proteomecentral.proteomexchange.org (accessed on 3 September 2021)) via the iProX partner repository with the dataset identifier PXD033183. The final data were analyzed and processed by Proteome Discoverer version 2.2 (Thermo Fisher from Thermo Fisher Technology Co., Ltd., Shanghai, China) software. Protein identification was performed using the UniProt rice database. The false positive rate of peptide identification was controlled below 1% and the threshold for the identification of differential proteins was FC > 1.2 or FC < 5/6 (*p* < 0.05). Protein function was annotated using gene ontology and KOG annotations (http://www.omicshare.com/tools (accessed on 15 September 2021)).

### 4.9. Validation Using Quantitative Real-Time PCR (qRT-PCR)

We first extracted RNA from rice anthers during meiosis. Nine DEPs were randomly selected for qRT-PCR analysis, including *BGIOSGA007251*, *BGIOSGA008908*, *BGIOSGA002684*, *BGIOSGA035762*, *BGIOSGA017916*, *BGIOSGA004771*, *BGIOSGA017346*, *BGIOSGA021194*, and *BGIOSGA027368*. Primer3 was used for primer design and all primers (listed in Appendix A) were synthesized by Sangon Biotech (Bioengineering Co., Ltd., Shanghai, China). We generated cDNA by reverse transcription of diploid rice and autotetraploid rice to be used as templates for downstream qRT-PCR experiments. Total RNA was isolated using the instructions of the RNA extraction kit (Qiagen from Kaijie Biological Engineering Co., Ltd., Shenzhen, China). cDNA templates were diluted to a uniform concentration, then qRT-PCR runs were performed using Hiscript^®^ III Supermix for qPCR kit (Nova). Actin was used as the internal reference gene for data standardization. All reactions were repeated three times, and the relative expression was then calculated using the 2−ΔΔCt method. PCR parameters were as follows: 20 μL reaction mixture, including SYBR Premix Ex Taq (Tli RNase H Plus from Takara Bio, Beijing, China) (10 μL), forward and reverse primers (0.8 μL at 10 mM), cDNA (30 ng/μL) 2.0 μL, and ddH_2_O 6.4 μL. The procedure consisted of initial denaturation at 95 °C for 5 min, followed by 45 cycles of denaturation at 95 °C for 10 s, annealing at 57 °C for 10 s, and extension at 72 °C for 15 s. The results were visualized using GraphPad Prism version 5.

## 5. Conclusions

Autopolyploidy, in which the added genomes come from the same species, is produced by direct doubling of diploid chromosomes. Compared with diploid rice, autotetraploid rice has lower seed setting rate. Researchers have been trying to overcome the main disadvantages of low seed setting rate and directly utilize the advantages of polyploids in grain quality, resistance, and biological yield to study and utilize autopolyploid rice [19,64]. In this study, the problem of low fertility of autotetraploid rice was explained by anther differences and proteomics analysis. In addition, abnormal chromosome segregation in autotetraploid rice was revealed by cytological dynamics. Therefore, this study provides a new idea for the study of anthers in the reproductive process of autotetraploid rice. In the future, chromosome behavior abnormalities in autotetraploid rice need to be further studied. At the same time, since the proteins related to tapetum are related to different regulatory mechanisms, researchers should also pay attention to them. With the deepening of our understanding, it is believed that the fertility problem of polyploid rice is finally solved, and polyploid rice breeding as a new breeding method will show attractive prospects.

## Figures and Tables

**Figure 1 plants-11-01647-f001:**
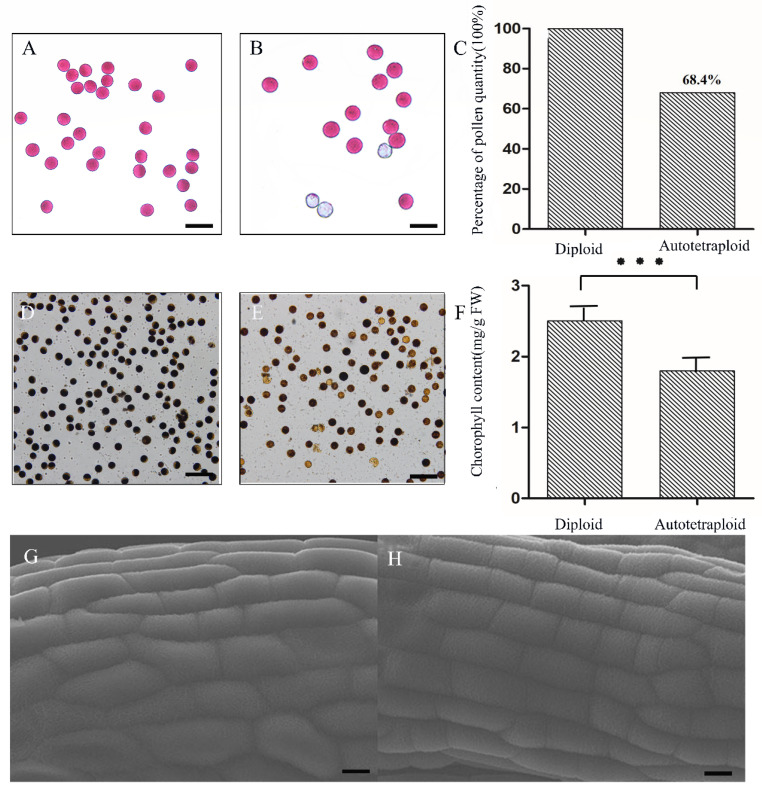
Fertility identification, chloroplast content determination, and anther ultrastructure observation. (**A**,**B**) Pollen grains subjected to Alexander staining; (**D**,**E**) pollen grains subjected to 1% iodine–potassium iodide staining. (**A**,**D**) Diploid; (**B**,**E**) autotetraploid; (**C**) bar charts showing percentage of mature pollen grains (number of mature pollen grains in diploid rice is taken to be 100%); (**F**) comparison of chlorophyll content, *n* = 6, *p* < 0.001 (The asterisk on the top indicates significant differences); (**G**,**H**) surface electron microscope images of anthers of diploid and autotetraploid rice during meiosis I; (**G**) diploid; (**H**) autotetraploid. Bar = 20 μm (**A**–**D**), 10 μm (**G**,**H**).

**Figure 2 plants-11-01647-f002:**
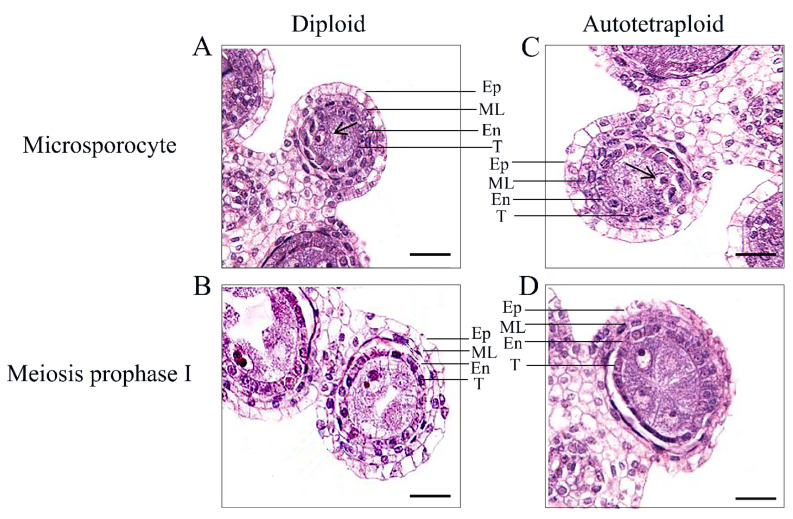
Observation on anther structure of diploid rice and autotetraploid rice. (**A**,**B**) Diploid; (**C**,**D**) autotetraploid, gradual degradation of tapetum in prophase I of meiosis. Note: Ep = epidermis layer; ML = middle layer; En = endothelium layer; T = tapetum layer. Bar = 100 μm.

**Figure 3 plants-11-01647-f003:**
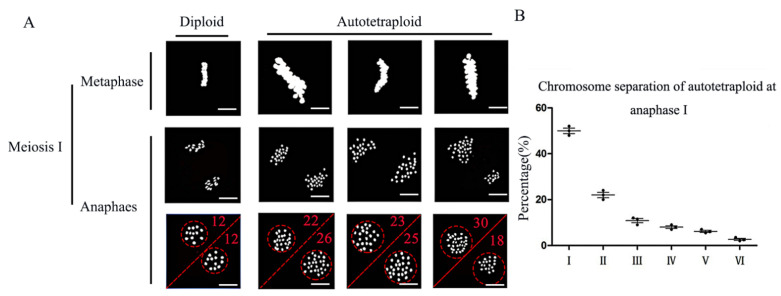
(**A**) Abnormal chromosome segregation behavior of autotetraploid rice at meiosis stage and (**B**) the percentage of the chromosome separation at anaphase I. Bar = 10 μm (I: 24/24; II: 23/25, 22/26, 21/27; III: 20/28, 19/29, 18/30; IV: 17/31, 16/32, 15/33; V: 14/34, 13/35, 12/36; VI: 11/37, 10/38).

**Figure 4 plants-11-01647-f004:**
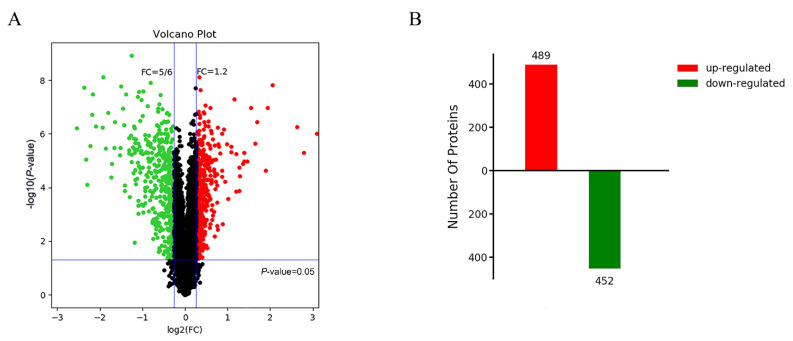
Screening and analysis of differential proteins. (**A**) Volcano map of differential protein distribution; (**B**) bar chart of up- and downregulated proteins.

**Figure 5 plants-11-01647-f005:**
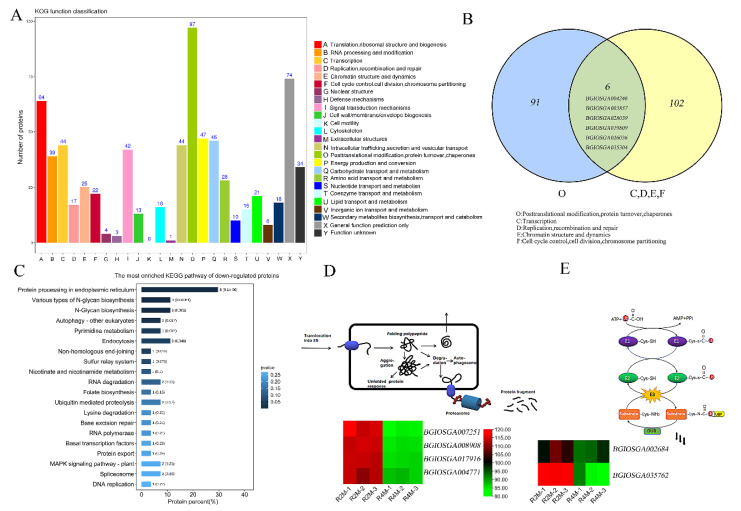
Correlation analysis between ERAD (ER–associated degradation) and protein ubiquitination. (**A**) KOG (Clusters of Orthologous Groups of proteins) functional classification of the differentially expressed proteins; (**B**) Venn diagram of O and C, D, E, and F; (**C**) KEGG (Kyoto Encyclopedia of Genes and Genomes) analysis of protein-degradation-related genes; (**D**) overview of the ERAD pathway; (**E**) overview of the protein ubiquitination process.

**Figure 6 plants-11-01647-f006:**
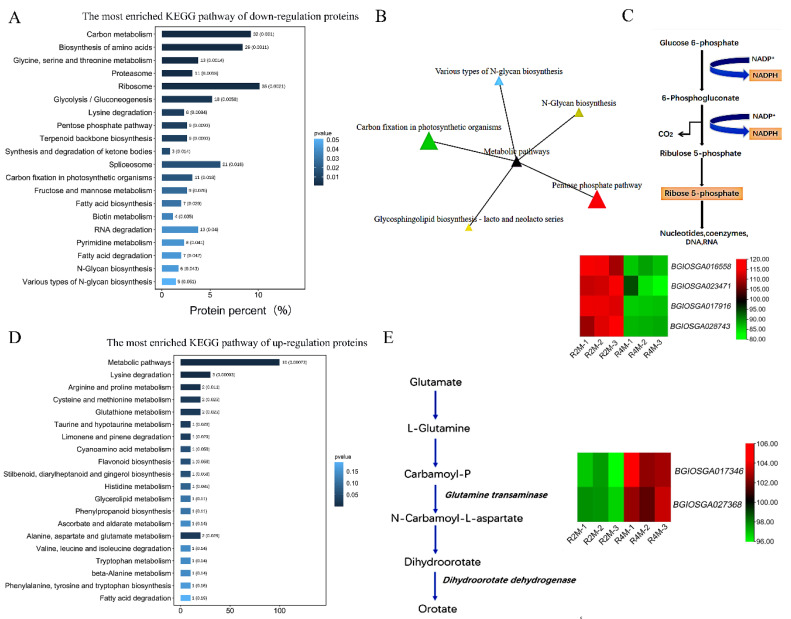
Correlation analysis of the pentose phosphate pathway (PPP) and glutamate decomposition. (**A**) KEGG (Kyoto Encyclopedia of Genes and Genomes) analysis of carbohydrate metabolism; (**B**) KEGG network diagram; (**C**) overview of the pentose phosphate pathway; (**D**) KEGG analysis of amino acid metabolism; (**E**) overview of glutamate decomposition.

**Figure 7 plants-11-01647-f007:**
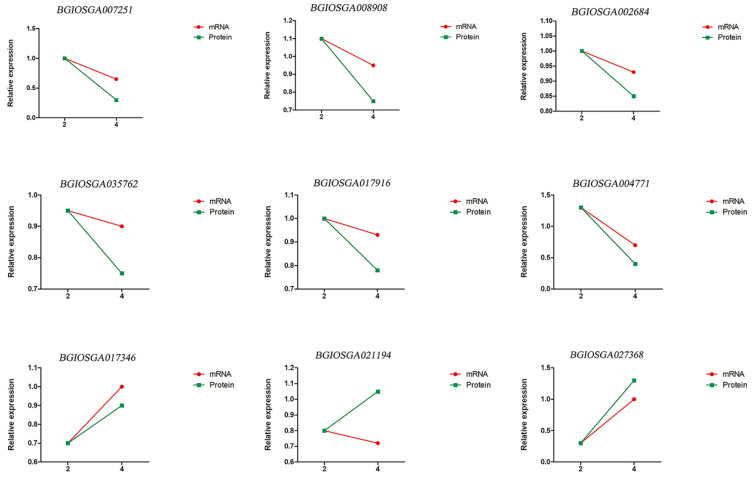
The qRT-PCR (real-time reverse transcription-PCR) validation of eight randomly selected differentially expressed proteins. The green columns represent relative protein expression and the red lines represent relative gene expression.

## Data Availability

The datasets generated and/or analyzed during the current study are available in the ProteomeXchange, http://proteomecentral.proteomexchange.org/cgi/GetDataset?ID=PXD033183 (accessed on 16 April 2022). All data are contained within this article and Appendix A.

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
