# Peer review of "Developmental Differences between Anthers of Diploid and Autotetraploid Rice at Meiosis"

_plants, 2022, doi:10.3390/plants11131647_

Round 1
Reviewer 1 Report
Dear editor,
this is a comparative article on the development of pollen grains in diploid and polyploid rice samples. The authors use different tools to show the paths of pollen inviability in polyploids. There are very good data, but I think the manuscript presentation should be more careful, especially in the discussion and conclusions. That is, in my assessment, this manuscript needs extensive revision.
I am not an expert in proteomics, so I suggest that you send the modified article to a new reviewer, more experienced in the subject.
Anyway, in the part where I dominate, such as meiosis and flow cytometry, I understand that better quality information and images are lacking.
Other details of my review are below.
Title:
Q1: Could the title be “Developmental differences between anthers of diploid and autotetraploid rice” ?
Abstract:
Q2, lines 11-13: Could the sentence be: “In this study, cytological analysis showed that at meiosis I stage, an unbalanced segregation of homologous chromosomes, as well as an early degeneration of tapetal cells in autotetraploid rice.” ?
Key words:
Q3: Please, put keywords in an alphabetical order.
Introduction:
Q4, lines 73-77: Could this sentence be: “An study in polyploid rice using whole staining transparent laser scanning confocal microscopy (WCLSM) showed a small polar anomaly, associated with embryo sac degeneration, and egg organ degeneration [26]. These authors proposed that abnormal embryo sacs would affect fertilization, reducing seed set. ?
Q5, lines 78-79: Which tissue are the authors referring to?
Q6, line 81: Mother cells of what? Please, improve the sentence.
Q7, lines 88-89: Please, note this sentence {Proteins, the final products of gene transcription, have the most direct impact on organisms.} In my humble opinion, the end product of gene transcription is a functional mRNA. A polypeptide chain should be the end product of translation and not necessarily a functional protein.
Q8, lines 92-93: This assumption is very controversial, since the variation in the sizes of plant genomes does not always culminate in failure, given that more than 70% of plant species are polyploid.
Results:
Q9, lines 132-133: Could the authors clarify this experimental strategy (S.E.M.) better?
Q10, lines 142-143: This sentence should appear right after the description of meiotic analysis. At this point in the text, there is no mention of the behavior of chromosomes in meiosis.
Q11, figure 2, page 5: The resolution of Fig.2 is very poor. It is very difficult to see details of anthers. Please, change by better images.
Q12, lines 157-158: It was not clear which meiotic processes were evaluated... All of which? Apparently, only the regularity of chromosomes, bivalent, univalent, and others was seen.
Q13, lines 161-162: Please, check this sentence {Therefore, we conclude that the meiosis process of autotetraploid rice was affected after polyploidization}. Authors could improve this sentence, because it looks like there was one more polyploidization event from the autotetraploid, and I believe this is not the case.
Q14, Figure 3: The image is too small and low resolution. This makes it difficult to count the associated or segregated chromosomes.
Q15, lines 180-184: Please, check this sentence {In addition, We also found DEPs classified into three categories related to “cell cycle control, division, chromosome partition,” “Replication, recombination and repair,” and “chromosome structure”; among them, cell cycle regulation is essential for pollen growth.}. Iit is necessary to define where (which tissue) the control of cell cycles is essential for the development of pollen grains. Note that the tapetum cells are degenerating at this point, and pollen mitoses are not cyclic, because cell divisions terminate in the formation of gametic cells. So, where are there cell cycles? In which anther tissues?
Q16, lines 171-211: I was left with a doubt about the scope of the results and the discussion in tissue terms. In addition to the connective, there are five tissues in the anthers of rice. As I understand it, the authors performed a proteomic analysis of complete anthers, comparing diploids and polyploids. So far so good. But many of the conclusions about the roles of some groups of proteins could not be attributed to one tissue or another, unless there was a more complete immunocytochemical analysis. How to know where exactly each set of proteins would be acting? Even though I have no expertise on the subject, this should be evident to me.
Discussion:
Q17: The first topic of the discussion looks like a repeat of the results rather than a typical discussion. I suggest that the discussion move away from mere description, to a model that addresses the biological/cytological dynamics of the process. Comparisons with ultrastructural studies (T.E.M.) would be welcome to discuss.
Q18, lines 285-324: The discussion is very focused on the ubiquitination process. Maybe that could show up in the title. Anyway, the discussion continues to look like a summary of the results. I prefer a discussion text that broadly compares the results with findings in other plant groups, trying not describe again what is already described in the results.
Q19: Not all results were discussed. If they are not important to the discussion, they should not appear in the article. Please verify.
Material and methods:
Q20, lines 337-344: Please provide more details about flow cytometry, such as patterns, number of events, the cytometer used, and others.
Q21, lines 361-362: Alexander staining shows viable cytoplasms at that exact time of staining, but this does not mean that they would remain viable. The best thing would be to associate pollen grain germination with this methodology. That way, your results would be more accurate.
Conclusions:
Q22: Conclusions again bring some repetitions of the results. This stretch should be shorter and more accurate, without redundancies.
Reviewer 2 Report
In the manuscript entitled: "Developmental Differences in Anthers at Meiosis Between Diploid and Autotetraploid Rice" (Number: plants-1765681), the Authors conducted a comparative study of the male lineage in the generative development of diploid and tetraploid rice. The formation of viable and fertile pollen grains is a multi-stage and complex process, and abnormalities can occur at any stage of the development of the male gametophyte. Therefore, comprehensive studies conducted by the authors, ranging from the morphology of the generative organ (anther), through microsporogenesis, and ending with gene expression, are solid. The research also has an application aspect, as it concerns the yield of an economically important plant, which is rice.
I found some of the problematic points in the manuscript, I only cite some of them below.
- line 133-136: “…involved in normal seed setting after the subsequent cracking of rice anthers…” – unclear sentence.
- line 146: “alexander staining;” – should be Alexander.
- line 153: Figure 2 – photos of the anther are of very poor quality; therefore, it is impossible to clearly define the stage of meiosis at the anther loculus. Besides, based on paraffin preparations and at such a low magnification, it cannot be said that: "... the tapetum layer has degraded." The presented image of tapetum cells (in Fig. 2B) may be the result of a preparation error, thus it should be corrected.
- line 171-172“According to the cytological results, the tapetum of autotetraploid rice developed abnormally at meiosis I stage, which affected pollen fertility.” – in my opinion, based on cytological observations, no such conclusion can be drawn.
- line 312: “…and causing abnormal tapetum development.” – you cannot write about "tapetum development" because during meiosis there is programmed death of tapetum cells.
- in the Discussion chapter in the last paragraph the Authors write again about "tapetum development" which is incorrect. In addition, they do not mention the chromosomal abnormal distribution that they observed during anaphase I, which is rather related to the incorrect of the karyokinetic spindle. The cause-and-effect sequence is also interesting. Which came first: abnormal PCD (programmed cell death) of tapetum cells - incorrect the karyokinetic spindle - abnormal chromosome segregation?
- chapter 4.4. 4.5. and 4.6 – in these chapters there are no citations concerning the techniques presented.
In conclusion, in my opinion, used the techniques are correct and the scientific value of the research presented in the manuscript is high.
Round 2
Reviewer 1 Report
Dear,
The authors accepted almost all the suggestions and the text was improved, however, the conclusions still look like a summary of the results. I suggest that the topic "Conclusions" be rewritten and shortened, bringing advances in knowledge on this topic, as well as future perspectives and their applications.
